# Prevalence of hepatitis B/C viruses and associated factors in key groups attending a health services institution in Colombia, 2019

Jaiberth Antonio Cardona-Arias[1]*, Juan Carlos Cataño Correa[2], Luis Felipe Higuita-Gutiérrez[1,3]

1 School of Microbiology, University of Antioquia, Medellín, Antioquia, Colombia, 2 Internal Medicine Infectious Diseases Section, University of Antioquia, Medellín, Antioquia, Colombia, 3 Faculty of Medicine, Cooperative University of Colombia, Medellín, Antioquia, Colombia

☉ These authors contributed equally to this work.
* jaiberth.cardona@udea.edu.co

**Data Availability Statement:** All relevant data are within the paper and its Supporting Information files.

## Abstract

Both hepatitis B virus (HBV) and hepatitis C virus (HCV) are major sources of morbidity and mortality worldwide; however, their prevalence in key groups in Colombia is not yet known. We aimed to analyse the prevalence of HBV and HCV and its associated factors in key groups who were treated at an institution providing health services in Colombia during 2019. This was a multiple-group ecological study that included 2,624 subjects from the general population, 1,100 men who have had sex with men (MSM), 1,061 homeless individuals, 380 sex workers, 260 vulnerable young people, 202 drug users, 41 inmates and 103 people from the lesbian, gay, bisexual and transgender community. Prevalence of infection with a 95% confidence interval and its associated factors was calculated for each group. Confounding variables were assessed using logistical regression and SPSS 25.0 software. Prevalence of HBV and HCV in the general population was 0.15% and 0.27%, respectively; 0.27% and 2.09% in MSM; 0.37% and 2.17% amongst homeless individuals; 0.26% and 0.0% amongst sex workers; 0.39% and 0.0% amongst vulnerable youth; and 5.94% and 45.54 amongst injecting drug users. In the multivariate HBV model, the explanatory variables included the study group, city of origin and the type of health affiliation; for HCV they were group, origin, sex, age group, health affiliation, use of drugs and hallucinogen use during sexual intercourse. A high prevalence of HBV and HCV were evidenced for both viral infections, which was, consequently, much higher within the key groups. The main associated factors that were identified related to origin and type of health affiliation and demonstrated a double vulnerability, that is, belonging to groups that are discriminated and excluded from many health policies and living under unfavourable socioeconomic conditions that prevent proper affiliation and health care.

## Introduction

Viral hepatitis, especially types B and C, which account for more than 95% of deaths in the population, generate a high morbidity and mortality burden worldwide, with figures higher to

**Funding:** Grant Gilead subvention 05239 "A call to action from the promotion of health and prevention of HIV-HBV-HCV infection" The funders had no role in study design, data collection and analysis, decision to publish, or preparation of the manuscript.

**Competing interests:** Gilead provided funding for this study. There are no patents, products in development or marketed products associated with this research to declare, and this does not alter our adherence to PLOS ONE policies on sharing data and materials.

those from tuberculosis or HIV/AIDS, but with less investment in diagnosis, prevention and treatment. The most concerning data are those related with hepatitis C virus (HCV) and hepatitis B virus (HBV). In the case of HCV, due to an increase in its morbidity with a worldwide prevalence of 1% [1,2], there are currently 75–85% of cases corresponding to chronic infection, 10–20% at risk for developing cirrhosis within 30 years and a 1–5% risk of hepatocellular carcinoma [3,4]. Furthermore, there are estimates of 1.75 million cases of HCV per year, however, due to low levels of screening most of the individuals infected are unaware of their serological status. This constitutes a serious clinical and public health issue due to an increased risk of transmission and progression to cirrhosis, hepatocellular carcinoma and death [1].

On the other hand and despite the existence of massive vaccination programmes, HBV continues to be a global public health issue [5–7] that accounts for two billion people that are serologically positive [8] and 325 million individuals with chronic infection [5]. It is the cause of chronic hepatitis, cirrhosis and hepatocellular carcinoma, the latter of which cause approximately 650 thousand deaths per year [6]. Additionally, around 8–16 million cases are estimated to be caused by blood transfusions mainly in Africa [9], decrease in vaccinations [5,8], escape mutants and viral mutations caused by selective pressure of treatments [6] and access barriers to control programmes [5], which further exacerbates this problem. The vaccine is available since 1992, and some of the population borne before that date in not vaccinated.

There are some studies on the prevalence of HBV and HCV infections in Colombia, and there is a lack of research on key groups, such as men who have sex with other men (MSM), homeless individuals, lesbians, transgender people, sex workers, injecting drug users and/or inmates. This was evident after applying the following syntax in Scielo *(ab*: *(hepatitis c)) AND (ab*: *(hepatitis b)) AND (colombia)* and the following syntax in PubMed *((hepatitis B [Title/ Abstract]) AND hepatitis C [Title/Abstract]) AND Colombia*. Despite this fact, several previous studies in the field are worth mentioning. An investigation with 619 subjects from four departments in Colombia, with Amerindian populations of Amazon River, female sex workers, female sex workers, doctors and nurses and nurses and dislocated people, found the following prevalence levels: HBsAg 5.66%, being statistically higher in Magdalena (8.39%) and without association to age or sex; anti-HBc was 28.43% with statistical differences in line with origin, sex and age, which were higher in men (34.36%), people older than 50 years (51.85%) and individuals from the Amazon (31.61%) [10]. A screening conducted on 3,369 subjects for HBV and 1,450 for HCV in a health service provider (IPS by its acronym in Spanish) in Medellín in symptomatic and asymptomatic people who had contact with an infected person (sexual partner or mother), and who according to medical criteria should be screened, found prevalence levels of 1.7% for the first round and 0.6% for the second round. Both were associated with age (higher in ages above 40 years) and HBV was found to present a higher incidence rate in men [11]. In 65,535 donors from the same city, the prevalence levels found for HCV and HBV were 0.0% and 0.1% respectively, the latter being statistically higher in men [12]. From the 39,825 donors studied from Montería, 0.32% had HBV and 0.04% had HCV, highlighting statistical association with age and sex, but unlike other studies, the highest proportion, in this case, was found in women [13].

In 302 donors recruited in a blood bank, HBsAg negative and Anti-HBc positive, the PCR shown 1.98% as occult HBV infection, subgenotype F3 [14]. In indigenous populations of the Colombian Amazon the prevalence of HBV (using anti-HBc) was 3.6% (46/1275) in children and 30.9% (177/572) in mothers; genetic studies in the positives cases showed a predominance of subgenotypes F1b and F1a, with identification of two HBV escape mutants G145R and W156* in a child with occult HBV; mutations L109R and G130E of the HBsAg sequence were also identified [15]. Specifically in people who inject drugs (PID), a study about HIV/HCV which included 50 people from Pereira-Colombia shown that had sex with other key groups

was the main risk factor [16], and a cross-sectional study with 918 PID from four Colombian cities (Armenia, Bogotá, Cúcuta and Pereira) found a prevalence of 27.3% in HCV, and the risk factors were: had a history of injection drug use of 5 years or more, higher injection frequency, daily use of gifted, sold, or rented needles and being HIV seropositive [17].

The prevalence of these studies is high and heterogeneous, and the available evidence suggests that, in key groups, this data might be even higher. Learning about the prevalence of both infections in different groups of the community is a public health priority in Colombia, as the country does not have active epidemiological surveillance programmes in place to combat these infections. Interestingly, these are treatable chronic infections, and their transmission can be controlled given the high efficacy of available antivirals [18].

The objective of this study was to compare the prevalence of HBV, HCV and associated factors in key groups who attended a health care provider (HCP) in Colombia in 2019.

## Materials and methods

### Type of study

Multiple-group ecological research study.

### Study population

A total of 5,771 individuals who attended the Fundación Antioqueña de Infectología (FAI) were included in the study in the period July to December 2019. The groups comprised individuals from the general population (2,624), MSM (1,100), homeless individuals (1,061), lesbians (15), transgender individuals (41), bisexuals (47), inmates (41), sex workers (380), vulnerable young people (260), and injecting drug users (202), who fulfilled the inclusion criteria of being members of one of the key groups of Colombian HIV control policies (with the exception of the general population group that served as a control), participating in intramural or extramural activities of the FAI for STI screening, and being 10 years of age or older (only were included adolescents and adults; after applying these criteria were excluded the individuals who did not sign the assent (for those under 18 years of age) or consent (from 18 years of age) informed, who demanded some type of remuneration to participate in the study, and those whose biological samples were not suitable for carrying out laboratory tests.

The concept of sample or sampling did not apply to the study because the entire population (institutional) was involved. *Stricto sensu* the results are representative of institutionalized populations (in Colombian HCP); however, the results can be extrapolated to groups that share the sociodemographic (Table 1) and sexual risk characteristics described for this population.

The participants were recruited by the FAI in intramural and extramural care campaigns in prisons, areas with high concentration of sex workers, homosocialisation areas and care centres for the homeless were included, from the main cities of Colombia, that is to say Bogotá (capital of Colombia and of the department of Cundinamarca), Medellín (capital of the department of Antioquia), Cali (capital of the department of Valle del Cauca), Quibdó (capital of the department of Chocó) y Pereira (capital of the department of Risaralda).

For the recruitment of the participants, the FAI carries out field work in the following phases: i) The FAI medical team presents the project and the advantages of performing STI screening to leaders of key groups, ii) The leaders of each group socialize the project, the proposal of the fieldwork and propose the sites for the application of the questionnaires and the sampling, iii) The FAI medical team performs the pre-test advice, explains and obtains the informed consent, provides the doctor's information and the place where the results will be delivered, iv) In positive cases, it is ensured that the participant gets care from their health service and treatment of the infection.

**Table 1. Distribution of the relevant demographic characteristics of the study population.**

| N 5.771 | | % | CI 95% |
|---|---|---|---|
| **Origin** | Medellín | 45.75 | 44.45–47.04 |
| | Cali | 20.93 | 18.87–21.99 |
| | Bogotá | 20.69 | 19.64–21.74 |
| | Quibdó | 1.65 | 1.31–1.98 |
| | Pereira | 1.21 | 0.92–1.50 |
| | Other | 9.77 | 8.99–10.55 |
| **Sex** | Woman | 41.93 | 40.65–43.22 |
| | Man | 58.07 | 56.78–59.35 |
| **Age group** | Adolescents (<21 years old) | 18.49 | 17.48–19.50 |
| | 21–40 years old | 52.82 | 51.52–54.11 |
| | 41–60 years old | 20.22 | 19.18–21.27 |
| | Over 60 years old | 8.47 | 7.75–9.20 |
| **Marital status** | Married/Common-Law Marriage | 47.95 | 46.65–49.24 |
| | Single | 51.71 | 50.41–53.00 |
| | Widow/Separated | 0.35 | 0.19–0.51 |
| **Education level** | None | 57.88 | 56.59–59.16 |
| | Primary | 4.49 | 3.95–5.03 |
| | Secondary | 27.50 | 26.34–28.66 |
| | Superior | 10.14 | 9.35–10.92 |

## Data collection

The study used primary sources of information based on a survey with sociodemographic and health data (S1 File) and application of diagnostic tests for both viruses. The survey items were selected based on the experience of the FAI and a review of the literature. The initial version of the instrument was subjected to an appearance validity process to ensure its applicability (according to the criteria of the medical and epidemiological team) and its acceptability (according to the criteria of the subjects of the study groups), with two physicians, two epidemiologists, an infectologist and five people from each study group. Because in this process of the face validation no generated changes in the instrument, the validity and relevance of its content to apply to the study population was confirmed. In its application, the survey was completed by the doctor and in some cases self- fill out by each participant according to your criterion.

HCV was detected with SD BIOLONE HCV, immunochromatographic rapid assay that detects antibodies in serum, plasma, or whole blood, by means of a strip coated with recombinant capture antigen with HCV proteins Core, NS3, NS4 and NS5. This test has a sensitivity of 100% and a specificity of 99.4%. The OnSite HBsAg Combo (Serum/Plasma/Whole Blood), a lateral flow immunoassay with 100% sensitivity and specificity, was used for HBV.

Selection and information biases were controlled through the application of the eligibility criteria by the medical team, standardisation of extramural work, information and motivation participation campaigns, pre and post-test counselling and the application of diagnostic tests whose risk of false results tended towards zero.

## Statistical analysis

Sociodemographic and health variables and risk factors in each group were described with relative frequencies (proportions). The prevalence of HCV and HCV in each study group was determined with a 95% confidence interval. The prevalence of both viruses was compared with

the sociodemographic and health variables and risk factors, with the Pearson's Chi square test. Variables that could have been confounding were identified, that is, those presenting an association with the prevalence of HBV or HCV and with another sociodemographic or health variable. A multivariate logistical regression model was performed with these variables for the HBV and another for the HCV, with the dual purpose of excluding confounders and identifying the explanatory variables of the prevalence of both in the study population. Dummy variables were constructed in the logistical regression models, defining the lowest prevalence as the reference group, and a Hosmer–Lemeshow goodness-of-fit test was used to evaluate model fit. Analyses were performed using SPSS 25.0 software with 95% confidence.

### Ethical aspects

The study was approved by the FAI Scientific Committee, who applied the guidelines of resolution 8430 of the Colombian Ministry of Health for research with vulnerable groups. The study was considered to have greater-than-minimal risk and the recommendations of the World Health Organisation (WHO) were to obtain informed consent and guarantee confidentiality, advice, quality of results and referral to health services for treatment- which were thoroughly followed.

## Results

Table 2 describes the sociodemographic and health characteristics of the study groups. There was a higher proportion of individuals from Medellín and Cali and a higher proportion of individuals who were under 40 years of age who had little or no schooling (except for vulnerable youth and inmates). There was also high heterogeneity regarding sex distribution, marital status, health affiliation and other health variables in the study groups.

In all of the study groups, more than 90% of subjects reported risky sexual intercourse. Risks with the lowest seroprevalence included biological accidents, sexual intercourse with individuals with sexually transmitted infections (STIs) and having an imprisoned sexual partner. The use of hallucinogens was higher amongst homeless individuals, having a new sexual partner was more frequent amongst MSM, vulnerable and bisexual youth and body piercings were more common in lesbians and bisexuals (Table 3).

Neither the prevalence of HBV, nor that of HCV presented statistical associations with marital status, receiving a blood transfusion or organ transplantation, receiving vaccination in the last year, identifying risky sexual relationships, having sexual relations with individuals diagnosed with STIs, changing sexual partners during the last six months, receiving money or psychoactive substances in exchange for money, being deprived of liberty (the study subject or his sexual partner), having body piercings, or biohazard accidents in the last year (Chi square p > 0.05).

The prevalence of HBV was statistically higher amongst injecting drug users (5.94%), in men (0.63%), in Afro-descendants (2.78%) and in people without health affiliation (1.11%). On the other hand, the prevalence of HCV was statistically higher amongst injecting drug users (45.54%), men (3.91%), people aged 21 to 30 years (3.85%) and 31 to 40 years (4.04%), uneducated individuals (3.68%), Caucasians (3.56%), those without health affiliations (4.29%), or those from the subsidised regime (4.31%), people who use hallucinogens during sexual intercourse (5.71%) and amongst those who have sex with people from key groups (4.56%) (Table 4).

Only three explanatory variables were identified in the multivariate HBV model: study group, city of origin and type of health affiliation. The probability for HBV in MSM, homeless individuals and vulnerable youth was five times higher than that found in the general

**Table 2. Percentage distribution (%) of the sociodemographic and health features of the study groups.**

| Feature | General | MSM | Homeless person | Sex worker | Vulnerable youth | Injecting drug user | Bisexual | Transsexual | Lesbian | Inmate |
|---|---|---|---|---|---|---|---|---|---|---|
| **N** | **2.624** | **1.100** | **1.061** | **380** | **260** | **202** | **47** | **41** | **15** | **41** |
| **Origin** | | | | | | | | | | |
| Medellín | 41.8 | 90.5 | 19.4 | 11.6 | 83.1 | 18.3 | 63.8 | 24.4 | 26.7 | 0.0 |
| Cali | 6.0 | 3.2 | 59.8 | 68.2 | 0.4 | 54.5 | 0.0 | 24.4 | 6.7 | 0.0 |
| Bogotá | 30.4 | 4.6 | 20.4 | 20.0 | 2.7 | 0.0 | 29.8 | 51.2 | 66.7 | 0.0 |
| Quibdó | 2.9 | 0.1 | 0.0 | 0.0 | 6.9 | 0.0 | 0.0 | 0.0 | 0.0 | 0.0 |
| Pereira | 18.8 | 1.6 | 0.4 | 0.3 | 6.9 | 27.2 | 6.4 | 0.0 | 0.0 | 100.0 |
| **Sex** | | | | | | | | | | |
| Woman | 58.5 | 12.6 | 14.1 | 94.7 | 57.3 | 17.3 | 38.3 | 36.6 | 93.3 | 12.2 |
| Man | 41.5 | 87.4 | 85.9 | 5.3 | 42.7 | 82.7 | 61.7 | 63.4 | 6.7 | 87.8 |
| **Age group** | | | | | | | | | | |
| Adolescents (< 21 years) | 19.6 | 24.6 | 8.5 | 6.3 | 46.9 | 11.9 | 31.9 | 2.4 | 26.7 | 2.4 |
| 21–30 years | 30.8 | 45.0 | 23.9 | 35.3 | 51.9 | 55.0 | 53.2 | 53.7 | 46.7 | 43.9 |
| 31–40 years | 16.0 | 16.1 | 23.7 | 29.5 | 0.0 | 26.7 | 12.8 | 24.4 | 20.0 | 19.5 |
| 41–50 years | 10.2 | 6.2 | 16.7 | 18.2 | 0.4 | 6.4 | 0.0 | 2.4 | 6.7 | 12.2 |
| 51–60 years | 9.1 | 6.4 | 19.7 | 8. | 0.8 | 0.0 | 2.1 | 9.8 | 0.0 | 12.2 |
| Older than 60 years | 14.3 | 1.7 | 7.5 | 2.1 | 0.0 | 0.0 | 0.0 | 7.3 | 0.0 | 9.8 |
| **Schooling** | | | | | | | | | | |
| None | 50.9 | 72.1 | 63.3 | 87.1 | 0.0 | 73.8 | 31.9 | 80.5 | 73.3 | 0.0 |
| Incomplete primary | 3.0 | 0.1 | 0.2 | 0.0 | 2.7 | 18.8 | 0.0 | 0.0 | 0.0 | 7.3 |
| Complete primary | 3.2 | 1.1 | 1.1 | 0.0 | 6.9 | 0.0 | 8.5 | 0.0 | 0.0 | 0.0 |
| Incomplete secondary | 0.9 | 0.3 | 10.6 | 1.8 | 1.5 | 2.0 | 2.1 | 0.0 | 0.0 | 9.8 |
| Completed secondary | 29.0 | 20.1 | 24.8 | 10.8 | 26.9 | 5.4 | 38.3 | 17.1 | 13.3 | 82.9 |
| Technical | 5.0 | 3.2 | 0.0 | 0.3 | 56.9 | 0.0 | 6.4 | 2.4 | 13.3 | 0.0 |
| University | 8.0 | 3.2 | 0.0 | 0.0 | 5.0 | 0.0 | 12.8 | 0.0 | 0.0 | 0.0 |
| **Marital status** | | | | | | | | | | |
| Married—Common Law Marriage | 41.1 | 69.9 | 49.4 | 67.4 | 1.5 | 43.1 | 19.1 | 70.7 | 46.7 | 9.8 |
| Single | 58.4 | 30.0 | 50.4 | 32.4 | 98.5 | 56.9 | 80.9 | 26.8 | 53.3 | 87.8 |
| Widowed—Separated | 0.5 | 0.1 | 0.2 | 0.3 | 0.0 | 0.0 | 0.0 | 2.4 | 0.0 | 2.4 |
| **Ethnicity (self-perceived)** | | | | | | | | | | |
| Afro-descendent | 0.4 | 0.3 | 1.8 | 0.3 | 0.4 | 0.0 | 2.1 | 0.0 | 0.0 | 0.0 |
| White | 50.0 | 71.6 | 63.1 | 83.7 | 0.0 | 69.8 | 31.9 | 80.5% | 73.3 | 0.0 |
| Mestizo | 49.6 | 28.1 | 35.2 | 16.1 | 99.6 | 30.2 | 66.0 | 19.5 | 26.7 | 100.0 |
| **Health affiliation** | | | | | | | | | | |
| Without affiliation | 6.0 | 11.9 | 20.9 | 15.8 | 4.6 | 12.9 | 17.0 | 14.6 | 13.3 | 14.6 |
| Subsidised regime | 34.3 | 26.8 | 52.2 | 53.2 | 32.7 | 53.5 | 8.5 | 63.4 | 13.3 | 17.1 |
| Contributory regime | 59.7 | 61.3 | 26.9 | 31.1 | 62.7 | 33.7 | 74.5 | 22.0 | 73.3 | 68.3 |
| **Other health features** | | | | | | | | | | |
| Hospitalisation in the last 12 months | 12.2 | 12.0 | 5.5 | 6.6 | 19.0 | 20.3 | 10.6 | 7.5 | 0.0 | 2.4 |
| Medication in the last month | 23.2 | 15.3 | 11.5 | 13.2 | 19.4 | 34.2 | 21.3 | 17.5 | 20.0 | 2.4 |
| Having received a transfusion or transplant | 7.7 | 5.9 | 2.5 | 2.6 | 11.9 | 5.9 | 8.5 | 5.0 | 0.0 | 2.4 |
| Having been vaccinated within the last year | 20.1 | 10.0 | 7.4 | 14.5 | 19.8 | 8.9 | 14.9 | 20.0 | 6.7 | 2.4 |

The table includes the percentages within each group (column percentages).

**Table 3. Percentage distribution (%) of sexual risk factors amongst the study groups.**

| Factor | General | MSM | Homeless person | Sex worker | Vulnerable youth | Injecting drug user | Bisexual | Transsexual | Lesbian | Inmate |
|---|---|---|---|---|---|---|---|---|---|---|
| Use of hallucinogens | 24.6 | 35.2 | 81.7 | 40.5 | 23.0 | 93.6 | 40.4 | 42.5 | 33.3 | 36.6 |
| Risky sexual behaviour | 96.7 | 96.8 | 99.9 | 100.0 | 90.8 | 100.0 | 91.5 | 97.6 | 100.0 | 100.0 |
| Intercourse with key group | 13.8 | 44.9 | 27.3 | 15.8 | 13.9 | 34.2 | 55.3 | 57.5 | 53.3 | 22.0 |
| Intercourse with people with STIs | 8.1 | 8.1 | 3.1 | 1.6 | 11.9 | 3.0 | 8.5 | 7.5 | 0.0 | 0.0 |
| New sexual partner within the last six (6) months | 35.4 | 52.5 | 26.2 | 40.3 | 55.6 | 30.7 | 59.6 | 40.0 | 33.3 | 56.1 |
| Having received psychoactive substances or money in exchange for sexual intercourse | 10.0 | 17.1 | 14.1 | 54.7 | 14.7 | 18.8 | 17.0 | 47.5 | 0.0 | 7.3 |
| In the last twelve (12) months the subject or his sexual partner has been deprived of liberty | 12.4 | 6.7 | 11.3 | 5.3 | 11.5 | 8.9 | 8.5 | 5.0 | 0.0 | Non-applicable |
| In the last twelve (12) months the subject or his sexual partner had piercings made | 23.2 | 22.3 | 8.6 | 11.1 | 25.4 | 13.4 | 31.9 | 10.0 | 46.7 | 19.5 |
| In the last twelve (12) months the subject has suffered biohazard accidents | 7.1 | 4.9 | 2.0 | 0.3 | 11.9 | 2.5 | 8.5 | 0.0 | 0.0 | 0.0 |

The table includes the percentages within each group (column percentages).

population; amongst sex workers it was 3.7 and it was 33.7 amongst injecting drug users. It was 3.5 amongst individuals without health affiliations compared to those from people affiliated with the contributory scheme. For HCV, the explanatory variables were group, origin, sex, age group, health affiliation and use of drugs and hallucinogens during sexual intercourse, with higher odds ratios in MSM (25.6), injecting drug users (77.5), inhabitants of Cali (17.4), men (2.1), people between 31 and 40 years of age (2.9), those affiliated with the subsidised regime (5.9) and those who use hallucinogens during sexual intercourse (Table 5).

## Discussion

Considering that there was a large population with subjects from different cities within the country, applying a test with high diagnostic validity (sensitivity 100% and specificity of 99.4%) resulted in the prevalence of HBV and HCV to be 0.15% and 0.27% amongst the general population, respectively. Said prevalence was 0.27% and 2.09% amongst MSM; 0.37% and 2.17% amongst homeless; 0.26% and 0.0% amongst sex workers; 0.39% and 0.0% amongst vulnerable youth; and 5.94% and 45.54% amongst injecting drug users. The explanatory variables for both infections study group were the origin and type of health affiliation. This is of great clinical and public health significance as it reveals the large number of people who are at risk for cirrhosis, carcinoma, and even death in the event of not receiving adequate treatment and follow-up for their condition, while also demonstrating a high risk of transmission of both viruses within the country.

HBV and HCV prevalence in the general population was 0.15% and 0.27%, which is lower than that reported worldwide (approximately 1%) and in Latin America (approximately 0.7%); that, despite not using the same type of test for HCV, it does allow comparison of figures, due to the low proportion of false positives (specificity 99.4%) or negatives (sensitivity 100%) in the tests used in this study, avoiding problems of over or under estimation of prevalence. This highlights an important improvement for this group, although these low frequencies, in the context of issues accessing treatment in Colombia, should draw attention to the presence of asymptomatic carriers who are at high risk of contracting cirrhosis or liver cancer [1,2].

**Table 4. Specific prevalence of HBV and HCV according to study group and sociodemographic and health characteristics.**

| Variable | Levels | N | HBV | | HCV | |
|---|---|---|---|---|---|---|
| | | | % (n) | CI 95% | % (n) | CI 95% |
| Group | General | 2.624 | 0.15 (4) | 0.04–0.39 | 0.27 (7) | 0.05–0.48 |
| | MSM | 1.100 | 0.27 (3) | 0.06–0.79 | 2.09 (23) | 1.20–2.98 [a*] |
| | Homeless | 1.061 | 0.37 (4) | 0.10–0.96 | 2.17 (23) | 1.24–3.09 [a*] |
| | Sex workers | 380 | 0.26 (1) | 0.01–1.46 | No cases | –– |
| | Vulnerable youth | 260 | 0.39 (1) | 0.01–2.12 | No cases | –– |
| | Injecting drug users | 202 | 5.94(12) | 2.43–9.45 [b**] | 45.54 (92) | 38.43–52.66 [b**] |
| Origin | Medellín | 2.640 | 0.15 (4) | 0.04–0.39 | 0.42 (11) | 0.15–0.68 |
| | Cali | 1.208 | 0.25 (3) | 0.05–.072 | 5.46 (66) | 4.14–6.79 [b**] |
| | Bogotá | 1.194 | 0.34 (4) | 0.09–0.86 | 0.50 (6) | 0.06–0.95 |
| | Others | 634 | 2.21 (14) | 0.98–3.43 [b**] | 9.78 (62) | 7.39–12.17 [b**] |
| Sex | Woman | 2.420 | 0.16 (4) | 0.05–0.42 | 0.58 (14) | 0.26–0.90 |
| | Man | 3.351 | 0.63 (21) | 0.35–0.91 [a*] | 3.91 (131) | 3.24–4.58 [b**] |
| Age group | Adolescents (< 21 years) | 1.067 | 0.09 (1) | 0.0–0.52 | 0.94 (10) | 0.31–1.56 |
| | 21–30 years | 2.008 | 0.50 (10) | 0.17–0.83 | 3.85 (77) | 2.97–4.70 [b**] |
| | 31–40 years | 1.040 | 0.63 (7) | 0.13–1.22 | 4.04 (42) | 2.79–5.28 [b**] |
| | 41–50 years | 603 | 0.33 (2) | 0.04–1.19 | 1.99 (12) | 0.79–3.19 [a*] |
| | 51–60 years | 564 | 0.71 (4) | 0.19–1.81 | 0.18 (1) | 0.00–0.98 |
| | Older than 60 years | 489 | 0.20 (1) | 0.00–1.13 | 0.61 (3) | 0.13–1.78 |
| Schooling | None | 3.340 | 0.63 (21) | 0.35–0.91 | 3.68 (123) | 3.03–4.34 [b**] |
| | Incomplete primary | 130 | No cases | –– | 5.38 (7) | 1.52–9.65 [b**] |
| | Complete primary | 129 | 0.77 (1) | 0.02–4.24 | No cases | –– |
| | Incomplete secondary | 158 | No cases | –– | 1.27 (2) | 0.15–1.50 |
| | Completed secondary | 1429 | 0.21 (3) | 0.04–0.61 | 0.77 (11) | 0.28–1.26 |
| | University | 263 | No cases | –– | 0.76 (2) | 0.09–2.72 |
| Ethnicity | Afro-descendent | 36 | 2.78 (1) | 0.07–14.53 [b**] | No cases | –– |
| | White | 3.286 | 0.55 (18) | 0.28–0.82 | 3.56 (117) | 2.91–4.21 [**] |
| | Mestizo | 2.449 | 0.25 (6) | 0.03–0.46 | 1.14 (28) | 0.70–1.59 |
| Health affiliation | Without affiliation | 630 | 1.11 (7) | 0.21–2.01 [a**] | 4.29 (27) | 2.63–5.95 [b**] |
| | Subsidised regime | 2.183 | 0.64 (14) | 0.28–1.00 | 4.31 (94) | 3.43–5.18 [b**] |
| | Contributory regime | 2.958 | 0.13 (4) | 0.04–0.35 | 0.81 (24) | 0.47–1.15 |
| Others | Hospitalisation | 627 | 0.48 (3) | 0.10–1.39 | 4.78 (30) | 3.03–6.54 [b**] |
| | Medication consumption | 1.081 | 0.46 (5) | 0.15–1.08 | 4.90 (53) | 3.57–6.24 [b**] |
| | Use of hallucinogens during sex | 2.342 | 0.77 (18) | 0.39–1.14 | 5.72 (134) | 4.76–6.68 [b**] |
| | Identified risk relationship | 5.619 | 0.43 (24) | 0.25–0.61 | 2.58 (145) | 2.16–3.00 [a*] |
| | Sex with key groups | 1.360 | 0.74 (10) | 0.24–1.23 | 4.56 (62) | 3.41–5.70 [b**] |

No cases of HBV or HCV were found in the following groups: bisexual, transgender, lesbian, inmates, residents of Quibdó, technical training.

[a] Statistically higher prevalence than the lowest prevalence subgroup.

[b] Statistically higher prevalence than all subgroups.

[*] $p < 0.05$

[**] $p < 0.01$.

Prevalence of HBV was 0.27% and that of HCV was 2.09% amongst MSM. Studies in other countries show varying results for HCV ranging from 1% in Peru [19] to 22.7% in Australia [20]. Although hepatitis B has long been considered a sexually transmitted infection, the role of sexual contact in the transmission of hepatitis C has become less clear [21]. In this sense, several studies have concluded that heterosexual transmission of HCV is inefficient, whereas

**Table 5. Multivariate logistic regression models to identify potentially explanatory factors for the prevalence of HBV and HCV.**

| | Wald | p | Odds ratio (CI 95%) |
|---|---|---|---|
| **HBV model variables** | | | |
| **Group** | **31.55** | **0.000** | |
| MSM/general population | 3.73 | 0.048 | 5.3 (1.00–28.30) |
| Homeless/general population | 4.09 | 0.043 | 5.1 (1.05–25.13) |
| Sex workers/general population | 1.18 | 0.277 | 3.7 (0.35–38.48) |
| Vulnerable youth/general population | 2.18 | 0.140 | 5.5 (0.57–52.32) |
| Injecting drug users/general population | 30.36 | 0.000 | 33.7 (9.65–117.86) |
| **Origin** | **20.24** | **0.000** | |
| Cali/Medellín | 0.05 | 0.823 | 0.8 (0.15–4.49) |
| Bogotá/Medellín | 2.16 | 0.141 | 3.3 (0.67–16.54) |
| Others/Medellín | 12.77 | 0.000 | 12.4 (3.11–49.10) |
| **Health affiliation** | **3.56** | **0.047** | |
| Unaffiliated/Contributory | 3.35 | 0.047 | 3.5 (1.00–13.39) |
| Subsidised/Contributory | 2.54 | 0.111 | 2.6 (0.81–8.20) |
| **HCV model variables** | | | |
| **Group** | **144.18** | **0.000** | |
| MSM/general population | 36.24 | 0.000 | 25.6 (8.95–73.70) |
| Homeless/general population | 4.46 | 0.035 | 3.1 (1.08–9.10) |
| Injecting drug users/general population | 71.70 | 0.000 | 77.5 (28.3–212.2) |
| **Origin** | **108.03** | **0.000** | |
| Cali/Medellín | 46.86 | 0.000 | 17.4 (7.70–39.3) |
| Bogotá/Medellín | 12.93 | 0.000 | 7.5 (2.43–22.21) |
| Others/Medellín | 104.28 | 0.000 | 98.3 (40.7–237.0) |
| **Sex** (Man/Woman) | **3.98** | **0.046** | 2.1 (1.01–4.34) |
| **Age group** | **9.83** | **0.048** | |
| <21 / >60 years | 0.11 | 0.736 | 1.3 (0.29–5.74) |
| 21–30 / >60 years | 2.43 | 0.049 | 2.8 (1.00–10.3) |
| 31–40 / >60 years | 2.52 | 0.048 | 2.9 (1.00–10.86) |
| 41–50 / >60 years | 0.77 | 0.380 | 1.9 (0.46–7.69) |
| 51–60 / >60 years | 0.83 | 0.364 | 0.3 (0.03–3.45) |
| **Health affiliation** | **34.41** | **0.000** | |
| Unaffiliated/Contributory | 6.79 | 0.009 | 2.9 (1.30–6.35) |
| Subsidised/Contributory | 33.99 | 0.000 | 5.9 (3.26–10.77) |
| **Medications** (Yes/No) | **10.38** | **0.001** | 2.4 (1.40–3.96) |
| **Use of hallucinogens during sex** (Yes/No) | **16.01** | **0.000** | 5.3 (2.33–11.90) |

other molecular epidemiological studies have identified HCV transmission clusters in MSM networks [22]. The causes behind increased sexual transmission of HCV in MSM are complex and result from the interaction of several factors, including unprotected anal intercourse, which represents a high risk of infection for the receptive partner. The practice of serosorting (participating in sexual activities with individuals with the same HIV status) has become increasingly common amongst MSM and constitutes an unsafe practice as it does not prevent other STIs, including HCV. Sexual practices such as *fisting* and sharing of sex toys can cause extreme dilation of anal tissue and micro-bleeding during sexual activity. Group sex practices can also cause injury to the mucosal surfaces and rectal bleeding. Recreational drug use during sexual intercourse generates disinhibition and increased sexual excitement and lowers the

perception of risk, which adds to the likelihood of contracting HIV and/or other STIs that may increase biological vulnerability [21,22]. In light of these facts, it is essential to promote safe sexual practices in this population as they generate significant impacts on infection not only by HBV and HCV, but also on HIV and other STIs.

Prevalence of HBV was 0.26% and 0.0% for HCV amongst sex workers. In a study carried out on sex workers in Brazil, the prevalence found for HBV was 17.1% and that for HCV was 0.7% [23]. In a meta-analysis of the prevalence of HCV in key populations in Latin America and the Caribbean, it was found that the frequency amongst sex workers was 2% [24]. Conversely, a study on sex workers in Vietnam found that the prevalence of HCV ranges between 8.8% and 30.4%; however, injecting drug use was the main cause of infection [25]. The few cases of HCV amongst sex workers and the high prevalence amongst injecting drug users, reaffirm the hypothesis of the low HCV transmission efficiency in heterosexual or homosexual female intercourse [26]. Further research must be carried out to explain the low prevalence of HBV and the difference with female sex workers from other countries in the region, such as Brazil with 17.1% [23], Argentina with 14.4% [27], Bolivia with 13.1% [28] and Venezuela with 13.8% [29], in which the design corresponds to seroepidemiological research (cross-sectional study), executed in population attended in health centers or nongovernmental organizations, like the present study.

Prevalence of HBV was 0.37% and that of HCV was 2.17% amongst homeless people. According to DANE (National Administrative Department of Statistics by its Spanish acronym) figures, there are approximately 22,790 homeless individuals in the main cities of Colombia. Studies that quantify both infections in this population are scarce; however, a study was found in California, United States which reported the prevalence of HBV as 1.17% and of 41.7% for HCV [30]. Research on HCV in this population was more prolific. In this sense, a 2012 meta-analysis reported that HCV prevalence ranges from 3.9% to 36.2% and combined prevalence of infection was 20.3% [31]. Another meta-analysis published in 2018 found a combined prevalence of 28% [32]. Discrepancies in results can be attributed to differences in sampling methods, diagnostic tests used (some studies use self-reporting techniques), criteria used to define the status of the homeless and inclusion of the determinants of infection. Beyond these differences between studies, it is recommended that marginalised groups undergo concurrent tests which, in addition to tracking HBV and HCV, can detect tuberculosis and HIV. In cases of infection, patients should be helped to overcome barriers to completing diagnostic exams and treatments and helped to plan for transportation, housing, nutrition and, where required, migration to their cities of origin.

The prevalence found for HBV was 0.39% and that of HCV was 0.0% amongst vulnerable youth. Vulnerable youth in Colombia include a population group that is more exposed to abuse of their fundamental rights, exclusion, poverty, inequality and various sorts of violence. Data on HBV prevalence amongst adolescents or young individuals are scarce and the figures on annual mortality are not yet known [33]. Compared to adults, there has been little focus on diagnosis and treatment within this population, partly because most patients are in the immune-tolerant phase and do not require treatment [33]. In this sense, it is necessary to carry out seroepidemiological studies stratified by age to assess the prevalence of HBsAg in different populations with estimates on the burden, morbidity, mortality, and need for treatment by region. Considering that vertical transmission (mother-to-child) and horizontal transmission during early childhood are the main routes of transmission of HBV and that they are responsible for most chronic infections, it is important to seek universal immunisation at birth and during childhood, mainly in vulnerable populations as an effective strategy for lowering the incidence of new infections; which becomes more relevant when taking into account the absence of data of vaccine programs in this population [33].

The prevalence of HBV amongst injecting drug users was 5.94% and that of HCV was 45.54%. Research on this topic is scarce in Latin American countries; however, a research study conducted in Brazil revealed that the prevalence of HCV in this population group is 35.5% [34]. A systematic review on the prevalence of these types of hepatitis amongst injecting drug users in European countries determined that national estimates of HBV ranged from 0.5% in Croatia, Hungary and Ireland to 6.3% in Portugal. For HCV, findings ranged from 13.8% in Malta to 84.3% in Portugal. Prevalence within this group was also reported to be higher than that found in the general population, men who have sex with men, inmates and migrants [35]. Another systematic review in Iran reported that the prevalence of HCV amongst injecting drug users was 47% [36]. Other authors have attributed the lower proportion of HBV infection by the availability of the vaccine (although the vaccination status against hepatitis is unknown in the study population) and because the risk of chronification of infection is relatively low when it is acquired during adolescence or adulthood [37]. The consistency of the findings between studies from different countries confirm that the population of injecting drug users may well be the group with the highest risk for HBV and HCV infection; therefore, they are proposed to be sentinel groups for monitoring infection and characterising circulating genotypes. Regarding measures to reduce infection, the establishment of centres for the distribution of disposable syringes, the performance of free diagnostic tests, the distribution of condoms and detoxification treatments such as the one carried out with methadone, are suggested.

The study group, the origin and the type of health affiliation were the explanatory variables for both infections. This information is key because all WHO member states have committed to a global reduction of hepatitis-related deaths by 65% and new infections by 90%, by 2030 [38]. In order to achieve this goal, it is necessary to focus on actions within the country by region and by special groups, such as injecting drug users or MSM and to eliminate barriers to universal and timely access to health services. Regarding the type of intervention, massive testing, universal vaccination for hepatitis B, improvement of access to antiviral treatments and reduction of their cost have been proposed. However, nationwide research studies are necessary to ensure evidence-based decision-making processes are taking place regarding the type of interventions that will provide the greatest public health benefits. In China, for example, investing in comprehensive HBV programming is expected to generate savings of more than $ 1.5 for every $ 1 spent by 2030 (38).

The limitations of this research include the temporal bias of the exposed statistical relationships, the absence of causal associations, and the non-discrimination between chronic and active infections, the lack of information about HBV and HCV prevalence in the country and some risk populations, and the fact of not being able to analyzed the samples using molecular techniques. Despite these limitations, the importance of prevalence studies should be highlighted as a key element for the initiation and/or orientation of all types of clinical, epidemiological, public health and research actions.

## Conclusion

The prevalence of both viral infections evidenced a differential level of risk in the study population was evident in the population, with much higher rates within the key groups. The main associated factors identified related to origin and type of health affiliation demonstrated a double vulnerability, that is, belonging to groups that are discriminated and excluded from various health policies and living under unfavourable socioeconomic conditions that prevent proper affiliation and health care. These results corroborate the need for improving active epidemiological surveillance; increasing resources for prevention, diagnosis and treatment; addressing

structural problems of health service delivery; promoting the inclusion of key groups in the management health policies; and improving the quantity and quality of scientific evidence in this field.

## Supporting information

**S1 File. Prevalence of viral hepatitis type B and C–survey.**
(DOCX)

## Acknowledgments

Walter Osorio de la Fundación Antioqueña de Infectología.

## Author Contributions

**Conceptualization:** Jaiberth Antonio Cardona-Arias, Luis Felipe Higuita-Gutiérrez.

**Data curation:** Jaiberth Antonio Cardona-Arias, Juan Carlos Cataño Correa.

**Formal analysis:** Jaiberth Antonio Cardona-Arias.

**Funding acquisition:** Jaiberth Antonio Cardona-Arias, Juan Carlos Cataño Correa, Luis Felipe Higuita-Gutiérrez.

**Investigation:** Jaiberth Antonio Cardona-Arias, Luis Felipe Higuita-Gutiérrez.

**Methodology:** Jaiberth Antonio Cardona-Arias.

**Project administration:** Jaiberth Antonio Cardona-Arias, Juan Carlos Cataño Correa.

**Resources:** Jaiberth Antonio Cardona-Arias, Juan Carlos Cataño Correa, Luis Felipe Higuita-Gutiérrez.

**Software:** Jaiberth Antonio Cardona-Arias.

**Supervision:** Jaiberth Antonio Cardona-Arias, Luis Felipe Higuita-Gutiérrez.

**Validation:** Jaiberth Antonio Cardona-Arias, Luis Felipe Higuita-Gutiérrez.

**Visualization:** Jaiberth Antonio Cardona-Arias, Luis Felipe Higuita-Gutiérrez.

**Writing – original draft:** Jaiberth Antonio Cardona-Arias, Juan Carlos Cataño Correa, Luis Felipe Higuita-Gutiérrez.

**Writing – review & editing:** Jaiberth Antonio Cardona-Arias, Luis Felipe Higuita-Gutiérrez.

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
