## [Decision Letter · Decision Letter 0]

24 Jun 2020

PONE-D-20-11570

Prevalence of hepatitis B/C viruses and associated factors in key groups attending a health services institution in Colombia, 2019

PLOS ONE

Dear Dr. Cardona-Arias,

Thank you for submitting your manuscript to PLOS ONE. After careful consideration, we feel that it has merit but does not fully meet PLOS ONE’s publication criteria as it currently stands. Therefore, we invite you to submit a revised version of the manuscript that addresses several points raised during the review process.

We look forward to receiving your revised manuscript.

Kind regards,

Isabelle Chemin, PhD

Academic Editor

PLOS ONE

Journal Requirements:

2. Please include additional information regarding the survey or questionnaire used in the study and ensure that you have provided sufficient details that others could replicate the analyses.

For instance, if you developed a questionnaire as part of this study and it is not under a copyright more restrictive than CC-BY, please include a copy, in both the original language and English, as Supporting Information.

3. We note you have included a table to which you do not refer in the text of your manuscript. Please ensure that you refer to Table 1 in your text; if accepted, production will need this reference to link the reader to the Table.

4. Please upload a copy of Supplementary Material 1 which you refer to in your text on page 8.

Reviewers' comments:

Reviewer's Responses to Questions

**Comments to the Author**

1. Is the manuscript technically sound, and do the data support the conclusions?

Reviewer #1: Partly

2. Has the statistical analysis been performed appropriately and rigorously? 

Reviewer #1: I Don't Know

3. Have the authors made all data underlying the findings in their manuscript fully available?

Reviewer #1: Yes

4. Is the manuscript presented in an intelligible fashion and written in standard English?

Reviewer #1: Yes

5. Review Comments to the Author

Reviewer #1: The study is interesting, however several studies and data of HBV and HCV infection in Colombian population are not included

The discussion of the data obtained has to be edited considering the technical limitation of the rapid test and also considering the results of the studies carried out in PID in different cities in Colombia

6. PLOS authors have the option to publish the peer review history of their article (what does this mean?). If published, this will include your full peer review and any attached files.

Reviewer #1: No

---

## [Author Response · Author response to Decision Letter 0]

28 Jul 2020

Medellín, July 21th 2020

Dra Isabelle Chemin

Academic Editor.

PLOS ONE

Manuscript Number PONE-D-20-11570 “Prevalence of hepatitis B/C viruses and associated factors in key groups attending a health services institution in Colombia, 2019”

Kind regards, 

Through this letter, we report the completion of all the changes suggested by the editors. The changes are highlighted in blue. Below we describe the changes realized, consistent with each reviewer suggestion.

Journal Requirements

Comment 1. Please ensure that your manuscript meets PLOS ONE's style requirements, including those for file naming. The PLOS ONE style templates can be found at

Answer: the change was made in the manuscript and the title sheet, according to the information of the journal.

Comment 2. Please include additional information regarding the survey or questionnaire used in the study and ensure that you have provided sufficient details that others could replicate the analyses. For instance, if you developed a questionnaire as part of this study and it is not under a copyright more restrictive than CC-BY, please include a copy, in both the original language and English, as Supporting Information.

Answer: the change was made, we attached the supplementary material 1, which includes the survey in English and Spanish. in addition, in “Data collection” we wrote:

The survey items were selected based on the experience of the FAI and a review of the literature. The initial version of the instrument was subjected to an appearance validity process to ensure its applicability (according to the criteria of the medical and epidemiological team) and its acceptability (according to the criteria of the subjects of the study groups), with two physicians, two epidemiologists, an infectologist and five people from each study group. Because in this process of the face evaluation no generated changes in the instrument, the validity and relevance of its content to apply to the study population was confirmed.

Comment 3. We note you have included a table to which you do not refer in the text of your manuscript. Please ensure that you refer to Table 1 in your text; if accepted, production will need this reference to link the reader to the Table.

Answer: the change was made, we referred the table 1 in the study population.

Comment 4. Please upload a copy of Supplementary Material 1 which you refer to in your text on page 8.

Answer: the change was made, we attached the supplementary material 1, which includes the survey in English and Spanish.

Reviewers' comments

Comment 1. Is the manuscript technically sound, and do the data support the conclusions? The manuscript must describe a technically sound piece of scientific research with data that supports the conclusions. Experiments must have been conducted rigorously, with appropriate controls, replication, and sample sizes. The conclusions must be drawn appropriately based on the data presented.

Reviewer #1: Partly.

Answer: To improve this aspect, the applied survey was added as supplementary material, in English and Spanish. The description of the way in which the survey was constructed, validated and applied was expanded.

Furthermore, in the introduction the available evidence (other studies) for Colombia was added, in the discussion the type of screening or diagnostic tests used in each study was described with more detail to improve comparability of results.

The conclusion was changed, eliminating the part that reported a high prevalence, which only applied to a study group (as the reviewer pointed out), in the new version it says: The prevalence of both viral infections evidenced a differential level of risk in the study population was evident in the population, with much higher rates within the key groups…

Comment 2. Has the statistical analysis been performed appropriately and rigorously?

Reviewer #1: I Don't Know.

Answer: In accordance with this type of epidemiological research, this study applied the statistical analyzes required to achieve the objectives; the statistical analysis is also adequate according to the type of variables measured (categorical). Such analyzes include:

• Sociodemographic and health variables and risk factors in each group were described with relative frequencies (proportions).

• The prevalence of HCV and HCV in each study group was determined with a 95% confidence interval.

• The prevalence of both viruses was compared with the sociodemographic and health variables and risk factors, with the Pearson's Chi square test.

• Variables that could have been confounding were identified with multivariate logistical regression.

• A multivariate logistical regression model was performed with the purpose of identifying the explanatory variables of the prevalence.

Comment 3. Have the authors made all data underlying the findings in their manuscript fully available? The PLOS Data policy requires authors to make all data underlying the findings described in their manuscript fully available without restriction, with rare exception (please refer to the Data Availability Statement in the manuscript PDF file). The data should be provided as part of the manuscript or its supporting information, or deposited to a public repository. For example, in addition to summary statistics, the data points behind means, medians and variance measures should be available. If there are restrictions on publicly sharing data—e.g. participant privacy or use of data from a third party—those must be specified.

Reviewer #1: Yes.

Answer: does not apply.

Comment 4. Is the manuscript presented in an intelligible fashion and written in standard English? PLOS ONE does not copyedit accepted manuscripts, so the language in submitted articles must be clear, correct, and unambiguous. Any typographical or grammatical errors should be corrected at revision, so please note any specific errors here.

Reviewer #1: Yes.

Answer: does not apply.

Comment 5. Review Comments to the Author. Please use the space provided to explain your answers to the questions above. You may also include additional comments for the author, including concerns about dual publication, research ethics, or publication ethics. (Please upload your review as an attachment if it exceeds 20,000 characters).

Reviewer #1: The study is interesting, however several studies and data of HBV and HCV infection in Colombian population are not included. The discussion of the data obtained has to be edited considering the technical limitation of the rapid test and also considering the results of the studies carried out in PID in different cities in Colombia.

Answer: the change was made, in the introduction the available evidence (other studies) for Colombia was added.

In the discussion we clarify that the detection test used in this study has sensitivity of 100% and specificity of 99.4% in HCV and 100% in HBV, which implies that false positive or negative results tend to zero. This implies that the prevalence is not under or over estimated, and therefore the comparison with other similar studies is pertinent.

Despite this clarification, this reviewer's comment was added in the limitations of the study, particularly the fact that we did not make diagnostic confirmation with molecular tests.

Comment 6. PLOS authors have the option to publish the peer review history of their article (what does this mean?). If published, this will include your full peer review and any attached files. Do you want your identity to be public for this peer review? For information about this choice, including consent withdrawal, please see our Privacy Policy.

Reviewer #1: No.

Answer: does not apply.

Answer: all the changes suggested by the reviewer on the article in pdf were made.

• In introduction: we change or delete some words, we add some considerations about vaccination and eligibility criteria of some studies, we clarify some data limited to Africa and we added some studies from Colombia.

• In discussion: we explain the data about the validity of the diagnostic tests, we add some details about the tests used in the studies cited in this section so that we can explain limitations in some comparisons of our results, we add several clarifications about the absence of data on vaccination programs in our study population, we added as a limitation not being able to apply detection using molecular tests, and the lack of information of HBV and HCV prevalence in the country and is some risk populations.

Despite the studies added to this version, it is important to clarify that in the first version of the manuscript we only intended to make it clear that there is heterogeneity, that there are few studies investigating both infections and that, in general, in Colombia there are no available many publications on this topic (as demonstrated by the search syntaxes that are explained in the introduction).

For this addition of studies, consistent with the reviewer's suggestion, several additional syntaxes were applied: (HBV [Title / Abstract]) AND (Colombia [Title / Abstract]) generated 39 results (only 25 in the last 10 years), and (HCV [Title / Abstract]) AND (Colombia [Title / Abstract]) 33 (only 22 in the last 10 years). ((HCV [Title / Abstract]) AND (people who inject drugs [Title / Abstract])) AND (Colombia [Title / Abstract]) only 3 results are generated. From the investigations carried out in Colombia, we not included those developed with people with HIV, hepatocellular carcinoma, effect of vaccination, genotyping, molecular characterization, abstracts of papers, topic reviews, among other research backgrounds that are not directly related to the topic of this research.

We appreciate your prompt evaluation and valuable comments that significantly improve the quality of our research.

We look forward to new suggestions.

Sincerely,

The authors.

---

## [Editor Report · Decision Letter 1]

21 Aug 2020

Prevalence of hepatitis B/C viruses and associated factors in key groups attending a health services institution in Colombia, 2019

PONE-D-20-11570R1

Dear Dr. Cardona Arias,

We’re pleased to inform you that your manuscript has been judged scientifically suitable for publication and will be formally accepted for publication once it meets all outstanding technical requirements.

Kind regards,

Isabelle Chemin, PhD

Academic Editor

PLOS ONE
---

## [Editor Report · Acceptance letter]

8 Sep 2020

PONE-D-20-11570R1 

Prevalence of hepatitis B/C viruses and associated factors in key groups attending a health services institution in Colombia, 2019 

Dear Dr. Cardona-Arias:

I'm pleased to inform you that your manuscript has been deemed suitable for publication in PLOS ONE. Congratulations! Your manuscript is now with our production department. 

Kind regards, 

on behalf of

Mrs Isabelle Chemin 

Academic Editor

PLOS ONE